# The Local Anaesthetic Procaine Prodrugs ProcCluster^®^ and Procaine Hydrochloride Impair SARS-CoV-2 Replication and Egress In Vitro

**DOI:** 10.3390/ijms241914584

**Published:** 2023-09-26

**Authors:** Clio Häring, Johannes Jungwirth, Josefine Schroeder, Bettina Löffler, Beatrice Engert, Christina Ehrhardt

**Affiliations:** 1Section of Experimental Virology, Institute of Medical Microbiology, Center for Molecular Biomedicine (CMB), Jena University Hospital, 07745 Jena, Germany; clio.haering@med.uni-jena.de (C.H.); johannes.jungwirth@med.uni-jena.de (J.J.); josephine.schroeder@med.uni-jena.de (J.S.); 2Institute of Medical Microbiology, Jena University Hospital, 07747 Jena, Germany; bettina.loeffler@uni-jena.de; 3Inflamed Pharma GmbH, 07745 Jena, Germany; b.engert@inflamedpharma.com

**Keywords:** SARS-CoV-2, local anaesthetics, procaine, antivirals

## Abstract

As vaccination efforts against SARS-CoV-2 progress in many countries, there is still an urgent need for efficient antiviral treatment strategies for those with severer disease courses, and lately, considerable efforts have been undertaken to repurpose existing drugs as antivirals. The local anaesthetic procaine has been investigated for antiviral properties against several viruses over the past decades. Here, we present data on the inhibitory effect of the procaine prodrugs ProcCluster^®^ and procaine hydrochloride on SARS-CoV-2 infection in vitro. Both procaine prodrugs limit SARS-CoV-2 progeny virus titres as well as reduce interferon and cytokine responses in a proportional manner to the virus load. The addition of procaine during the early stages of the SARS-CoV-2 replication cycle in a cell culture first limits the production of subgenomic RNA transcripts, and later affects the replication of the viral genomic RNA. Interestingly, procaine additionally exerts a prominent effect on SARS-CoV-2 progeny virus release when added late during the replication cycle, when viral RNA production and protein production are already largely completed.

## 1. Introduction

Severe acute respiratory syndrome coronavirus 2 (SARS-CoV-2) is a positive-sense single-stranded RNA virus with a genome of around 30 kb [1]. The enveloped virus particles carry the spike protein on their surface with which they bind their cellular receptor angiotensin converting enzyme 2 (ACE2) [2,3,4]. The spike protein is primed by either transmembrane protease serin 2 (TMPRSS2) or Cathepsin L [5]. The virus particles are taken into the host cell via endocytosis or fuse directly with the plasma membrane [5,6]. Once the viral genome is released into the cytoplasm, cellular ribosomes produce two large polyproteins containing the 16 non-structural proteins of SARS-CoV-2 from open reading frame (ORF)1a and ORF1ab, which make up the first two thirds of the genome. These polyproteins need to be cleaved by the viral proteases known as 3C-like protease and papain-like protease. The non-structural proteins then establish the viral replication–transcription complex [7]. This includes modifying the intracellular membrane structures of the endoplasmic reticulum (ER) to form double-membrane vesicles (DMVs) in which replication and transcription can take place [8,9]. This likely serves to hide the replication process from cellular defence mechanisms [10]. During replication, full-length negative-sense copies of the viral genome are produced that serve as templates for the production of new positive-sense genome copies. These are used for the transcription of new genomes and the translation of more polyproteins and are integrated into new virus particles [11]. The translation of the viral structural proteins spike (S), membrane protein (M), envelope (E) and nucleocapsid (N) as well as several accessory proteins first requires a modified form of transcription [12,13,14]. A set of subgenomic negative-sense RNAs is produced from the genome via discontinuous transcription from which subgenomic positive-sense RNAs (sgRNAs) are then generated [15]. These sgRNAs consist of a 5′ untranslated sequence, which is the same 5′ sequence as that of the full-length genome, followed by the gene sequence for the specific structural or accessory protein, skipping ORF1 and all other previous ORFs [13]. Translation via cellular ribosomes occurs in the cytoplasm for the full-length genome as well as for sgRNAs. The lipid envelope of new virus particles is formed by budding into the ER-Golgi intermediate compartment (ERGIC). New virus particles are released via exocytosis using non-standard exocytosis pathways such as lysosomal exocytosis [16,17].

The hallmarks of severe coronavirus disease 2019 (COVID-19), caused by SARS-CoV-2, are pneumonia, pulmonary oedema, acute respiratory distress syndrome and multiple organ failure, probably due to viremia [18,19]. Many patients show cardiovascular complications accompanied by vascular abnormalities, high inflammatory markers and markers of cell disturbance [20,21]. These data point to massive problems in the regulation of innate immune responses [22,23,24].

Vaccination is the most effective method of protection against severe disease. However, vaccinations do not confer the same protection against newly emerging virus subtypes [25,26], and some persons cannot or will not be vaccinated. Accordingly, the identification of effective pharmaceutic ingredients is needed for the treatment of infected persons. There are several direct acting antivirals that are currently in use for the treatment of COVID-19 that target different stages of viral replication, namely remdesivier, molnupiravir, favipiravir (RNA polymerase inhibitors) and nirmatrelvir (protease inhibitor) [27,28,29,30,31]. Besides targeting the virus itself, novel therapeutic strategies inhibit virus-supportive cellular factors, or excessive immune responses, which result in tissue damage and inflammation [32,33]. To this end, extensive efforts have been made to identify the host protein interactions of SARS-CoV-2 to discover new potential drug targets [34,35]. Repurposing existing drugs that are used to treat other diseases against such targets can offer a way to quickly develop new treatment options.

Here, we investigate if procaine, an ester-type local anaesthetic, represents a potential candidate for further antiviral studies. Local anaesthetics are described to exhibit anti-inflammatory and antioxidative properties [36,37,38,39,40,41]. Others report regulatory functions of local anaesthetics on cellular factors, such as G-protein-coupled receptors (GPCRs) [42], and mitogen-activated protein kinases (MAPKs) [43]. Notably, GPCRs and MAPKs mediate virus-induced antiviral cellular functions, but are also used by viruses for the purpose of ensuring efficient replication [44,45,46]. A few studies further indicate the antiviral potential of procaine against different viruses, such as herpes simplex virus (HSV) and West Nile virus (WNV) [47,48], mainly by blocking virus entry into the cell.

We investigate the procaine prodrugs ProcCluster^®^ (also known as Procainum-hydrogen-carbonate) and procaine hydrochloride for their antiviral effects on SARS-CoV-2 infection in vitro.

## 2. Results

### 2.1. ProcCluster^®^ and Procaine Hydrochloride Treatment Results in Reduced Replication Efficacy of SARS-CoV-2 In Vitro

To determine a range of non-toxic concentrations of procaine for further experiments, MTT-based proliferation assays on Calu-3 cells were performed to exclude any effects of ProcCluster^®^ (PC) or procaine hydrochloride (PHCl) on cell viability. The results indicate that both PC and PHCl in concentrations of up to 5 mM do not notably affect proliferation and viability in this human lung carcinoma cell line (Figure 1a).

Next, we aimed to determine an effective concentration 50% (EC_50_) for both substances (Figure 1b). Calu-3 cells were infected with the SARS-CoV-2 isolate SARS-CoV-2/hu/Germany/Jena-vi005159/2020 (B1.1 linage, early variant) or SARS-CoV-2/hu/Germany/Jena-0114749/2021 (AY.126 linage, delta variant) and treated with PC and PHCl in concentrations of up to 4 mM. Plaque assays were used to analyse viral titres after 24 h of infection. Both substances showed a clear inhibitory effect on SARS-CoV-2 titres. PC and PHCl exhibited EC_50_ values that were very similar to each other in the low millimolar range. Notably, the EC_50_ values for both substances were lower when used against the SARS-CoV-2 delta variant compared to the early SARS-CoV-2 variant, specifically 1.23 mM (95% confidence interval (CI) 0.91–1.60) compared to 0.49 mM (95% CI 0.35–0.6) for PC and 1.04 mM (95% CI 0.88–1.21) compared to 0.51 mM (95% CI 0.46–0.57) for PHCl. An inhibition of viral titres using similar concentrations of PC and PHCl could also be observed in infected Vero-76 cells (Appendix A).

The inhibitory effect of procaine treatment on viral replication could further be seen in the Western blot analysis of the viral spike protein. Spike protein expression in Calu-3 cells infected with the B1.1 SARS-CoV-2 variant for 24 h was reduced in a concentration-dependent manner when the cells were treated with PC or PHCl (Figure 1c).

### 2.2. The SARS-CoV-2-Induced Interferon and Cytokine Response Is Reduced in Procaine-Treated Cells

Since one hallmark of severe COVID-19 is enhanced cytokine expression accompanied with detrimental inflammation and cell death, the impact of both substances on SARS-CoV-2-mediated cytokine and chemokine expression as well as viral mRNA synthesis were investigated via a quantitative real-time PCR (qRT-PCR) analysis at 24 h post infection (p.i.) (Figure 2a). As expected, treatment with 2.5 mM procaine resulted in reduced levels of SARS-CoV-2 RNA (N1 gene) compared to the untreated infected cells. The mRNA synthesis of the cytokines interleukin-6 (IL-6), tumour necrosis factor alpha (TNFα) and interferon-gamma-induced protein 10 kDa (IP-10) as well as the interferons (IFNs) IFNλ1, IFNλ2/3 and IFNβ was reduced in the presence of procaine in comparison to the untreated infected cells. The expressions of the antiviral mediators interferon-induced GTP-binding protein MxA and 2′-5′-oligoadenylate synthetase 1 (OAS1) were increased in the SARS-CoV-2-infected cells compared to the uninfected cells but procaine treatment did not result in a reduction.

Similar results were observed with the flow cytometry analysis of cell supernatants 24 h p.i. (Figure 2b). As seen on the RNA level, the amount of IFNβ, IFNλ1, IFNλ2/3, TNFα as well as IP-10 was increased in the untreated SARS-CoV-2-infected cells (though the increase was not as pronounced for TNFα) and was, in comparison, reduced in the procaine-treated infected cells. The levels of IFNγ and IFNα2 were not increased by SARS-CoV-2 infection in these cells. The release of new virus particles was measured as early as 8 h p.i. and more strongly at 12 h p.i. [49], which corresponds well with previous reports of the duration of SARS-CoV replication cycles [50,51,52]. Since cytokine measurements were performed on cells infected for 24 h, corresponding to more than one replication cycle, the observed effects on the mRNA and protein levels are likely related to the overall reduced infection.

### 2.3. ProcCluster^®^ Treatment Inhibits a Late Stage of the SARS-CoV-2 Replication Cycle

Host-directed antiviral therapies could potentially target any or even several steps in the viral replication cycle. Since procaine clearly inhibited SARS-CoV-2 replication in the 24 h multicycle experiments, its mode of action in single-cycle experiments was investigated next. A Western blot analysis at 8 h p.i. shows a reduced cellular spike content in the early SARS-CoV-2 variant as well as the delta-variant-infected Calu-3 cells treated with 2.5 mM PC or PHCl (Figure 3a). The effect was more pronounced with the SARS-CoV-2 delta variant, which agrees with the lower EC_50_ values (Figure 1b).

Single-cycle time-of-addition experiments with infected cells were conducted next (Figure 3b,c). The treatment of Calu-3 cells with PC starting 30 min prior to infection or immediately during infection had no added inhibitory effect on the progeny virus titres compared to when the PC treatment was started at 2 h p.i. This indicates that the SARS-CoV-2 uptake into the cell is not affected by procaine treatment. Accordingly, treatment with procaine only prior to infection or only during infection had no effect on the progeny virus titres (Figure 3b). We additionally investigated the mRNA expression of ACE2 and TMPRSS2 in infected as well as uninfected Calu-3 cells after 24 h of procaine or solvent treatment. The expressions of both ACE2 and TMPRSS2 were not affected by infection or procaine treatment (Appendix A). Remarkably, treatment with 2.5 mM PC or PHCl starting 6 h p.i. had the same inhibitory effect (approx. 90% reduction) on the viral titres (measured at 12 h p.i.) as when treatment was started 2 h p.i. (Figure 3c). The inhibitory effect decreased slightly when the procaine treatment was started at 9 h p.i., but progeny virus production was still inhibited by 60 to 70%. At 9 h p.i., the release of viral particles from the infected cells already begun, which could explain the increase in the viral titres seen when procaine was added 9 h p.i. compared to when it was added 6 h p.i. It is therefore especially remarkable that procaine treatment nonetheless inhibited progeny virus production when added 9 h p.i. To confirm this, the cells were infected with 3 MOI of SARS-CoV-2 and left untreated up until 9 h p.i., after which the medium was collected for plaque assays and replaced with fresh medium containing PC, PHCl or solvent. The fresh medium was then collected again after 3 h (12 h p.i. overall). As expected, the titres in all the samples were the same prior to treatment at 9 h p.i. (dark grey bars in Figure 3d). The virus titres obtained after 3 h under treatment, on the other hand, were significantly reduced in the procaine-treated samples compared to the control (light grey bars in Figure 3d). This indicates that procaine is capable of inhibiting a very late stage of the SARS-CoV-2 replication cycle and limiting the production of new viral particles even after progeny virus release has already begun. Such late action of procaine does not notably explain the reduced spike protein load observed in the Western blot analysis of the 8 h samples described earlier, which indicated additional effects earlier in the replication cycle.

### 2.4. ProcCluster^®^ Treatment Differentially Affects SARS-CoV-2 Replication and Transcription

To investigate the potential effects of procaine at earlier times during infection, we infected cells with the early SARS-CoV-2 variant and added 2.5 mM PC at 2 h after infection. The RNA from the cell lysates collected at 4 and 8 h p.i. was then analysed via qRT-PCR. We originally used primer pairs amplifying regions coding for N1 and spike protein. These primer pairs would amplify their target region from the full-length genome (gRNA) as well as any subgenomic copies and therefore do not offer any information on the effects of procaine on the production of gRNA and sgRNAs specifically (Figure 4a). Interestingly, in both cases, the RNA levels were reduced in the PC-treated samples compared to the control at 4 h as well as 8 h after infection (Figure 4b). We then used two more types of primer sets to differentiate between the gRNA and sgRNA copies. The first set contained primers for regions of the genome coding for non-structural protein 3 (nsp3) and the main subunit of the viral RNA-dependent RNA polymerase (RdRp/nsp12). These parts of the viral genome are not transcribed as sgRNAs and therefore purely represent the replication of the full-length viral genome. The RNA levels for nsp3 and the RdRp were not or only slightly reduced at 4 h p.i., while at 8 h p.i., they were significantly reduced (Figure 4c). The second set of primer pairs all share the same forward primer that was previously described by Dagatto et al. [53]. This primer binds within the 5′ lead sequence of the viral genome, while the respective reverse primers target parts of the structural protein genes. In the gRNA, these sequences are several thousand bases apart and will not produce a PCR product. In the subgenomic copies of the genome, however, the lead sequence comes into close range with the respective structural protein gene (Figure 4a). These primer pairs can therefore be used to investigate the levels of sgRNA copies specifically. Interestingly, unlike the full-length copies at 4 h p.i., the levels of the sgRNA copies of the E, spike, N1 and M genes were already significantly reduced in the PC-treated samples compared to the control and continued to be inhibited at 8 h p.i. (Figure 4d). Procaine therefore seems to affect the transcription of the viral genome earlier than it affects the replication of the full-length genome.

### 2.5. Procaine Treatment Inhibits Phospholipase A_2_

Several previous publications reported that procaine affects the activity of phospholipase A_2_ (PLA_2_), although the size and directionality of the effect is dependent on the concentration used and the source of the PLA_2_ [54,55,56,57,58]. The enzyme is responsible for the conversion of membrane phospholipids with two fatty acid tails into lysophospholipids with one fatty acid tail and a free fatty acid. We used a commercially available fluorescent PLA_2_ substrate (Thermo Fisher, Waltham, MA, USA) to investigate the effects of PC and PHCl on the activity of PLA_2_ from Calu-3 cells. A PLA_2_-containing cell lysate was generated by lysing Calu-3 cells with distilled water for 30 min. Both PC and PHCl showed a concentration-dependent inhibition of PLA_2_ activity in the cell lysate (Figure 5a). At a procaine concentration of 2.5 mM, PLA_2_ activity was inhibited by approximately 40%. The activity of PLA_2_ from bee venom was not affected by procaine treatment (Appendix A). Since PLA_2_ acts on membrane phospholipids and changes their overall shape, it influences the composition and structure of the external membrane as well as the internal membrane structures [59]. The inhibition of PLA_2_ could therefore influence the extensive membrane remodelling that is required for the formation of the SARS-CoV-2 replication organelles, impacting replication and transcription. As the envelope of SARS-CoV-2 is derived from internal cellular membranes, this could also alter the membrane structure of progeny virus particles. To investigate whether procaine treatment influences the heat stability of the new viral particles produced, we subjected SARS-CoV-2 containing samples from PC, PHCl or solvent-treated Calu-3 cells to heat treatment at 50 °C for 20, 40, 60 and 80 min (Figure 5b). The virus titres decreased from 5- to 10-fold for every 20 min at 50 °C but there was no significant difference in the heat lability between the virus particles derived from the control cells and procaine-treated cells.

Additionally, the potential effects of procaine on the main SARS-CoV-2 protease, 3C-like protease (3CL), were investigated, as a previous publication reported that procaine inhibited the activity of the protease trypsin [60]. PC treatment led to only a small inhibitory effect on the 3CL protease at 2.5 mM, while PHCl treatment did not cause any effect (Figure 5c). Considering the pronounced effects of both PC and PHCl on viral replication described above, it seems unlikely that the small inhibitory effect of PC on the 3CL protease contributes significantly to the above findings.

## 3. Discussion

Developing host-directed therapies for COVID-19 as potential treatment options could offer new advantages, since host-directed therapies are less likely to cause viral resistance development and could offer additional beneficial effects beyond the inhibition of viral replication. Here, we studied whether the local anaesthetic procaine shows antiviral potential in vitro for further investigation as a host-directed therapy against COVID-19. Our data show that procaine in low millimolar concentrations can cause a strong reduction in SARS-CoV-2 replication (Figure 1). This was accompanied by a reduction in the SARS-CoV-2-induced cytokine and IFN response on the mRNA as well as the protein level (Figure 2). The hyperactivation of the immune system is a prominent problem in patients with severe or even fatal COVID-19. Therefore, anti-inflammatory effects would be a beneficial secondary effect [61]. Although procaine is known to have anti-inflammatory effects on its own, the effect on cytokine expression that was observed here is likely at least partly related to the reduced viral titres in response to procaine treatment. A relationship between viral titres and cytokine as well as IFN levels has also been observed in COVID-19 patients [62].

Several studies on human patients with SARS-CoV, MERS-CoV and SARS-CoV-2 indicate that there is a limited or delayed induction of the type-I and type-III IFN responses and that this has negative consequences as it allows for increased viral loads [63,64,65], while others have shown a robust activation of the IFN response in COVID-19 patients [66]. We observed a strong induction of IFNβ, IFNλ1 and IFNλ2/3 in response to SARS-CoV-2 infection in our cell culture model (Figure 2), and a similar induction was observed in other in vitro models [67]. These differences are likely due to the very different complexities of the complete human immune system and a single-cell-type cell culture model. It is nonetheless important to note that procaine treatment limited the interferon response of these epithelial cells in proportion to the virus load. A potentially harmful independent early IFN response limiting effect was not observed. IFNα2 and IFNγ, on the other hand, were not induced by SARS-CoV-2 infection in our experiments (Figure 2b). The induction of these IFNs in COVID-19 patients varies on an individual basis and depends on disease severity [68]. Interestingly, although procaine treatment resulted in a reduced SARS-CoV-2-induced IFN response as discussed, there was no effect on the interferon-regulated factors MxA and OAS1 (Figure 2a), which are important for an antiviral state of the cells. An upregulation of OAS1 and MxA despite little induction of the interferon response was also observed in COVID-19 patients [65], and similar results were reported on a three-dimensional human alveolar stem cell model [69].

Although procaine was originally developed and used as a local anaesthetic agent, it was investigated for its potential antiviral functions early on. One study showed an antiviral effect of procaine against HSV and WNV that was mediated by the inhibition of viral entry into the cell [47]. In contrast, our experiments show that procaine did not inhibit the entry of SARS-CoV-2 into the cell in the concentrations employed here (Figure 3). Procaine treatment also did not affect the expression of the most prominent entry factors ACE2 and TMPRSS2 in Calu-3 cells (Appendix A). The most pronounced limitation of HSV and WNV entry in the aforementioned publication was observed after a 15 min pre-treatment with around 70 mM of procaine. The effect of such high concentrations on SARS-CoV-2 virus entry was not examined here due to the toxic effects that these concentrations would have in longer treatments. We could, however, show that procaine effects the synthesis of new viral RNA (Figure 4) and additionally exerts an effect late in the replication cycle around assembly and egression that limits progeny virus production (Figure 3d).

The specific detection of sgRNAs via qRT-PCR has been used to detect the presence of an actively replicating virus in several studies involving patient samples and animal models [70]. We used gRNA- and sgRNA-specific primers here to show that the transcription of sgRNAs is inhibited early on (4 h p.i.) by procaine, while the replication of the gRNA is only affected later (8 h p.i.) (Figure 4). Treatment was started 2 h p.i. in our experiments. A recent study using single-molecule FISH probes determined that there is little active replication and transcription of SARS-CoV-2 at 2 h p.i. Replication and transcription then progress over the next hours, with the sgRNA/gRNA ratio increasing until around 8–10 h p.i. [71]. The fact that procaine affects sgRNA synthesis but not gRNA synthesis early on suggests that the effect on gRNA synthesis might be a secondary effect of the inhibition of sgRNA synthesis. Alternatively, this pattern could also reflect a later onset of large-scale replication compared to transcription [71], with the more active process of transcription being affected more noticeably by procaine treatment. The observed inhibition of viral replication and transcription could be caused by a direct effect on these processes but might just as well be caused by more indirect effects of procaine on the cellular environment.

The fact that procaine treatment causes a strong reduction in viral titres even when added 9 h p.i. (Figure 3d) indicates a second potentially independent mechanism by which procaine limits progeny virus production. SARS-CoV-2 is assembled by budding into the ERGIC [72]. How SARS-CoV-2 then travels to the plasma membrane to be released by exocytosis is comparatively poorly understood [73], but some studies suggest that it transports to the plasma membrane via lysosomes [16,17,74] and/or recycling endosomes [75], while others could not definitively confirm this [76]. A virus egress inhibiting effect of procaine even after large amounts of viral protein and viral RNA have been produced indicates that it affects either the budding of the virus or the transport of newly formed virus particles to the plasma membrane.

Interestingly, the inhibitory effect of procaine in multi-cycle experiments was more pronounced in the delta variant (AY.126) than the early SARS-CoV-2 variant (B1.1). The delta variant carries several mutations in its spike protein, such as L452R and T478K, allowing for more efficient binding to the ACE2 receptor and improved immune escape compared to earlier variants. A mutation in the furin cleavage site of the spike protein, P681R, additionally improves the efficiency of spike subunit cleavage [77,78]. These mutations lead to an increased replication efficacy of the SARS-CoV-2 delta variants in humans as well as in cell cultures. We currently have no indication that procaine treatment has any significant direct effects on SARS-CoV-2 virus proteins. A host-directed mechanism of action is therefore more likely. The stronger inhibition of the SARS-CoV-2 delta variant’s replication is therefore probably due to differences in the replication kinetics in vitro that effect the relative potency of the inhibitor treatment.

In addition to the inhibitory effects on SARS-CoV-2 infection, procaine also inhibited the activity of PLA_2_ (Figure 5a). There are several different known groups of PLA_2_ enzymes that vary in the composition of their active site, in their need for Ca^2+^ for their catalytic activity and in their localisation within (or outside of) the cell. There is therefore still a need for further investigation into which subgroups of PLA_2_ enzymes are specifically affected by procaine treatment. Cytosolic PLA_2_ has been reported to be involved in several different membrane-trafficking processes [59] and has been shown to play a role in the replication cycle of several viruses. Upon the infection of the cell, SARS-CoV-2 causes extensive remodelling of cellular organelles, including the remodelling of the ER to form double membrane vesicles as replication compartments [72,79,80]. This seems to involve the activity of cytosolic PLA_2_. PLA_2_ activity is increased in coronavirus-infected cells, lysophospholipids produced by PLA_2_ are present in the membranes of DMVs and inhibiting cytosolic PLA_2_ subsequently reduces the formation of DMVs and limits coronavirus replication [81]. The inhibition of calcium-independent PLA_2_ has been shown to reduce the budding of vesicles from the membrane of Caco-2 cells and negatively affects the budding of influenza A virus [82]. The inhibition of PLA_2_ could therefore reasonably play a role in the inhibitory actions of procaine described above. A further investigation of the activity of PLA_2_ in living and infected cells will be required in the future to potentially confirm these theories. PLA_2_ enzymes are notably also responsible for the release of arachidonic acid from membrane phospholipids and are therefore important players in inflammatory processes involving eicosanoids and leukotrienes [83]. The modification of such processes by procaine would likely be relevant in infections of more complex systems.

The EC_50_ of procaine treatment determined here was between 0.5 and 1.2 mM depending on the SARS-CoV-2 variant, which corresponds to around 180 µg/mL and 430 µg/mL, respectively, for PC or 140 µg/mL and 330 µg/mL for PHCl. Studies on the pharmacokinetics of procaine have found steady-state serum concentrations of up to 20–60 µg/mL to be tolerable in humans after procaine infusion [84,85]. The concentrations used here are therefore comparatively high. Information on the serum concentrations that can safely be achieved via the administration of PC is currently not yet available. Further investigation on the kinetics of the more stable and bioavailable PC would be required to better contextualise the concentrations used here.

Overall, procaine effectively limits the production of SARS-CoV-2 at two different points in the replication cycle, inhibiting virus replication as well as release, and warranting further studies of its antiviral mechanisms and potential applications.

## 4. Materials and Methods

### 4.1. Cell Culture and Virus Cultivation

Calu-3 cells (ATCC cat. no. HTB-55; provided from the laboratory of Stephan Ludwig, Münster, Germany) and Vero-76 cells (ATCC cat. no. 1587; cell stock of the former Institute for Virology and Antiviral Therapy, Jena, Germany) were cultivated in DMEM (Gibco, Waltham, MA, USA) supplemented with 10% foetal calf serum (FCS; PAN-Biotech, Aidenbach, Germany). The SARS-CoV-2 isolates SARS-CoV-2/hu/Germany/Jena-vi005159/2020 [MW633322.1] (early variant containing a D614G mutation; nextstrain clade 20B; pango linage B1.1) [49] and SARS-CoV-2/hu/Germany/Jena-0114749/2021 [ON650061] (delta variant; nextstrain clade 21J; pango linage AY.126) [86] were cultured on Vero-76 cells in DMEM with 10% FCS.

### 4.2. Pharmacological Substances

ProcCluster^®^ (PC) is a patented substance (PCT/EP2018/074089; EP21157974.3), also known as Procainum-hydrogen-carbonate (ProcHHCO_3_ * NaCl), which was manufactured and supplied by inflamed pharma GmbH (Jena, Germany). PC is based on procaine and is stabilised by the mineral salt. The clustering process, which incorporates the active ingredient procaine into a protective shell, allows for the use of procaine—which was previously only available for parenteral use—for oral and dermal administration. PC has been produced in a GMP-compliant manner since 2008 and is used for the production of prescription medicines. Procaine hydrochloride (PHCl) (Caesar & Loretz GmbH, Hilden, Germany) is a prodrug of procaine.

### 4.3. MTT Assay

Calu-3 cells were seeded into 96-well plates 24 h prior to the experiment. The cells were treated with solvent (H_2_O) or treated with a serial 1:2 dilution of PC or PHCl for 24 h, after which 25 µL of 5 mg/mL MTT (Sigma Aldrich, St. Louis, MO, USA) was added, and cells were incubated for a further 2 h. The supernatant was carefully removed, and DMSO was added to lyse the cells. Metabolic activity was determined via relative absorbance at 562 nm in comparison to the solvent-treated control using a FLUOstar Omega plate reader (BMG Labtech, Ortenberg, Germany).

### 4.4. EC_50_ Determination

Calu-3 cells were seeded into 12-well plates the day prior to the experiment. Cells were washed and infected with SARS-CoV-2 with a multiplicity of infection (MOI) of 0.5 in DMEM (10% FCS) and simultaneously treated with PC or PHCl at the indicated concentrations. The cells were washed again with PBS after 2 h, supplied with fresh DMEM (10% FCS) supplemented with PC or PHCl and incubated at 37 °C and 5% CO_2_ for 24 h, after which virus titres were analysed via plaque assay. Viral titres of treated samples relative to those of the untreated samples (100%) were used to determine EC_50_ values via nonlinear fit (least squares; variable slope) (GraphPad Prism 9.3.1).

### 4.5. Infection of Cell Culture

Calu-3 or Vero-76 cells were seeded 36 h or 24 h prior to the experiments, respectively. Infection was carried out by adding SARS-CoV-2 stock diluted in DMEM (10% FCS) to the cells for 2 h. Cells were washed with PBS afterwards and incubated with DMEM (10% FCS) for the indicated times with PC or PHCl added where indicated.

### 4.6. Plaque Assay

Vero-76 cells were seeded into 6-well plates to create a confluent cell layer. The cells were infected with serial dilutions of a sample in PBS/BA (0.2% BSA, 1 mM MgCl_2_ and 0.9 mM CaCl_2_) supplemented with 100 U/mL Pen/Strep for 90 min. The inoculum was then replaced with MEM supplemented with 0.2% BSA, 0.01% DEAE Dextran (Pharmacia Biotech, Uppsala, Sweden), 0.2% NaHCO_3_ (Biozym, Rosemead, CA, USA), 100 U/mL Pen/Strep and 0.9% agar (Oxoid, Wesel, Germany) and incubated at 37 °C and 5% CO_2_ for 3 days. Plaques were visualised using neutral red staining.

### 4.7. qRT-PCR

RNA from Calu-3 cells was isolated using the RNeasy Mini Kit (Qiagen, Hilden, Germany) according to the manufacturer’s instructions. The QuantiNova Reverse Transcription Kit (Qiagen, Hilden, Germany) (Figure 2) or QuantiTect Reverse Transcription Kit (Qiagen, Hilden, Germany) (Figure 4) was used for cDNA synthesis from 400 ng of total RNA. For qRT-PCR, the QuantiNova SYBR Green PCR Kit (Qiagen, Hilden, Germany) was used. Cycle conditions were set as follows: 95 °C for 2 min, followed by 40 cycles of 95 °C for 5 s and 60 °C for 10 s. The qPCR cycle was ended by a stepwise temperature-increase from 60 °C to 95 °C (1 °C every 5 s). All primers used are listed in Table 1. Subgenomic RNA was detected using the “SARS-CoV-2 lead fw” primer in combination with a reverse primer for the specific gene in question.

### 4.8. Cytokine Measurements

Cytokine content of cell culture supernatants was determined via flow cytometry using the Legendplex™ human antivirus response panel kit (740349) (BioLegend, San Diego, CA, USA).

### 4.9. Western Blot

For Western blot analysis, cells were lysed with Triton lysis buffer (TLB; 20 mM Tris-HCl, pH 7.4; 137 mM NaCl; 10% Glycerol; 1% Triton X-100; 2 mM EDTA; 50 mM sodium glycerophosphate; 20 mM sodium pyrophosphate; 5 μg/mL aprotinin; 5 μg/mL leupeptin; 1 mM sodium vanadate; 0.2 mM Pefabloc and 5 mM benzamidine) for 30 min at 4 °C. Cell lysates were centrifuged, and 5× Laemmli buffer (10% SDS, 50% glycerol, 25% 2-mercaptoethanol, 0.02% bromophenol blue, 312 mM Tris, 6.8 pH) was added (1:5 dilution) to the supernatant. The mixture was boiled at 95 °C for 10 min. Samples were subjected to SDS-PAGE and Western blot analysis. SARS-CoV-2 spike protein was detected using a rabbit polyclonal anti-SARS-CoV-2 spike S2 antibody (#40590-T62, Sino Biological, Hong Kong, China). Equal protein load was verified using a mouse monoclonal anti-ERK2 antibody (sc-1647, Santa Cruz, CA, USA) or rabbit anti-ERK1/2 antibody (#4695, Cell Signaling Technology, Danvers, MA, USA). Quantification was performed using Fiji (Image J. JS v0.5.7; https://ij.imjoy.io). Samples were normalised using the loading control.

### 4.10. Phospholipase A2 Assay

PLA_2_ activity of Calu-3 cell lysates was determined using the EnzCheck^TM^ Phospholipase A2 Assay Kit by Invitrogen (Thermo Fisher Scientific, Waltham, MA, USA) according to the manufacturer’s instructions. Confluent Calu-3 cells were scraped from the cell culture dish in cold PBS using a rubber policeman. Cells were pelleted via centrifugation and resuspended in distilled water for 30 min on ice. After 30 min, equal volumes of 2× reaction buffer (supplied by the kit) were added, and samples were centrifuged again. The supernatant was used as sample containing PLA_2_ in the assay. Fluorescence was measured using a FLUOstar Omega plate reader (BMG Labtech, Ortenberg, Germany) (excitation: 485 nm; emission: 520 nm).

### 4.11. 3CL Protease Assay

Activity of 3CL protease was measured using the EnzCheck^TM^ 3C-like Protease Assay Kit by Invitrogen (Thermo Fisher Scientific, Waltham, MA, USA) according to the manufacturer’s instructions. Fluorescence measurements were performed using a FLUOstar Omega plate reader (BMG Labtech, Ortenberg, Germany) (excitation: 355 nm; emission: 460 nm).

### 4.12. Software

All statistical analyses and curve fitting were performed using GraphPad Prism Version 9.3.1. Western blot quantification was performed using Fiji (Image J. JS v0.5.7; https://ij.imjoy.io).

## Figures and Tables

**Figure 1 ijms-24-14584-f001:**
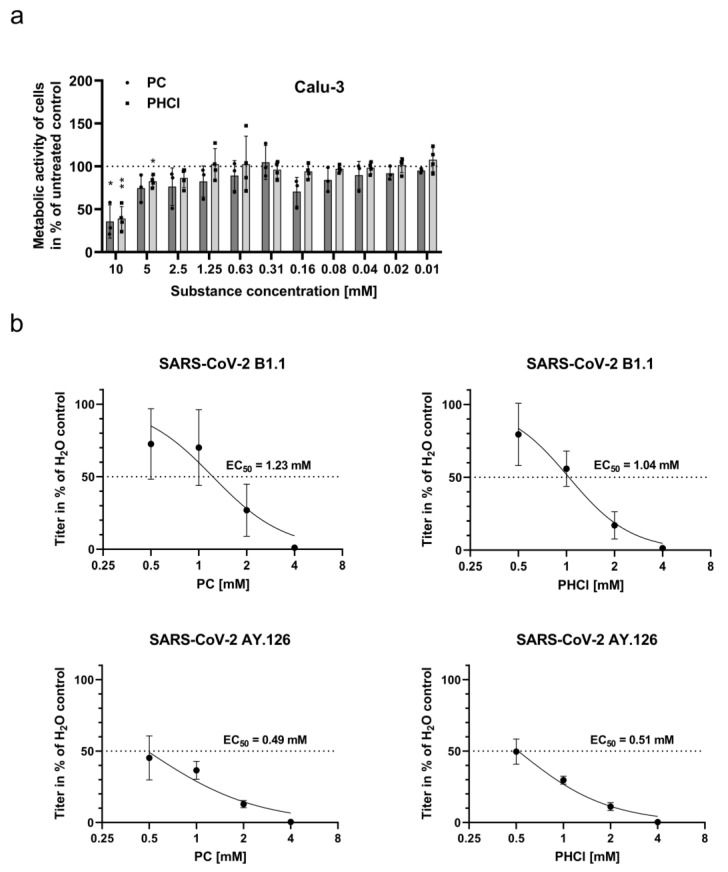
PC and PHCl treatment results in inhibition of SARS-CoV-2 replication at non-toxic concentrations in vitro. (**a**) Calu-3 cells were treated with the indicated concentrations of PC, PHCl or solvent (H_2_O) for 24 h, after which cell viability was determined via MTT assay. Values are given in percent of the solvent-treated control. Statistical significance was determined using one-sample t-tests comparing all means to 100%: * *p* < 0.05; ** *p* < 0.01. (**b**) Calu-3 cells were infected with 0.5 MOI of the SARS-CoV-2 variants (early variant B1.1 or delta variant AY.126) in the presence of PC, PHCl or H_2_O for 2 h. The inhibitor or solvent was again added after a washing step, and viral titres in the supernatant were determined via plaque assay at 24 h post infection (p.i.). The virus titres of solvent-treated infected cells were arbitrarily set to 100%. The means (±SD) of three independent experiments including two biological samples are shown. EC_50_ values were calculated using a 4-parameter nonlinear least squares fit in GraphPad Prism. (**c**) Calu-3 cells were pre-incubated with the indicated concentrations of PC or PHCl for 30 min prior to infection with 0.5 MOI of SARS-CoV-2 B1.1 variant. The cells were treated with the inhibitors during infection and again after a media change 2 h p.i. The cells were lysed 24 h p.i. and used for SDS-PAGE and Western blot analysis. Representative images of three independent experiments are depicted. Spike protein expression relative to the loading control was quantified using ImageJ.

**Figure 2 ijms-24-14584-f002:**
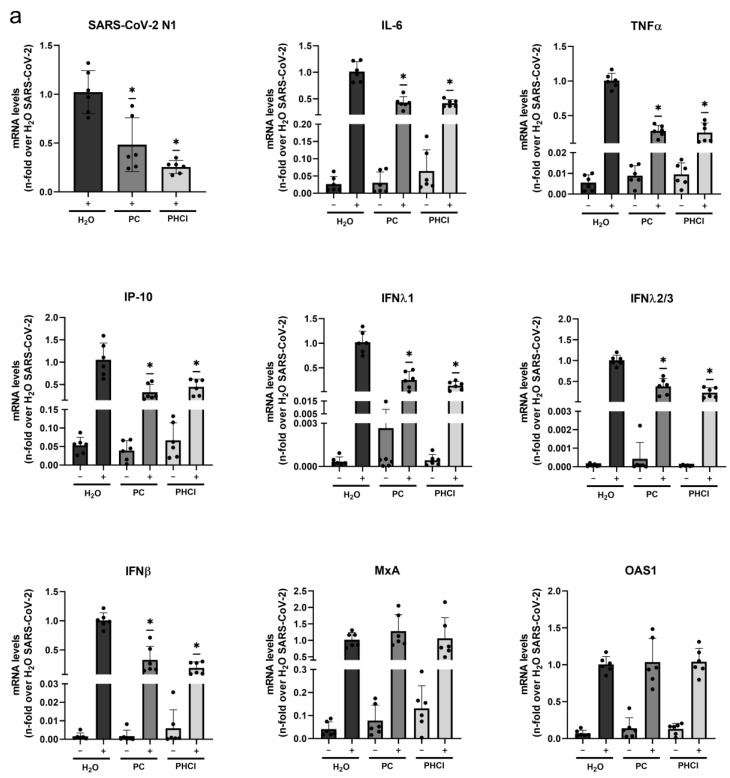
PC and PHCl treatment cause a reduction in SARS-CoV-2-induced IFN and cytokine levels. Calu-3 cells were pre-treated with 2.5 mM PC or PHCl or solvent control for 30 min prior to infection and then infected with 0.5 MOI of the early SARS-CoV-2 variant for 2 h in the presence of the inhibitor. The inhibitor was again added after a medium change, and the cells were further incubated to 24 h p.i. (**a**) RNA was extracted, and the replication of SARS-CoV-2 (detecting the N1 region) as well as the transcription of the indicated cytokines and chemokines was quantified via qRT-PCR. Solvent-treated infected samples were arbitrarily set to 1. The mean (+SD) of three independent experiments with biological duplicates is depicted (black dots represent individual values). Statistical significance was determined using one-sample Wilcoxon test and compared to 1; * *p* < 0.05. (**b**) The amount of the indicated cytokines in the supernatants was determined using flow cytometry. The mean (+SD) of three independent experiments with biological duplicates is depicted. Solvent-treated infected samples were arbitrarily set to 100%. Statistical significance was determined using one-sample *t*-test and compared to 100%: * *p* < 0.05; ** *p* < 0.01; *** *p* < 0.001; **** *p* < 0.0001.

**Figure 3 ijms-24-14584-f003:**
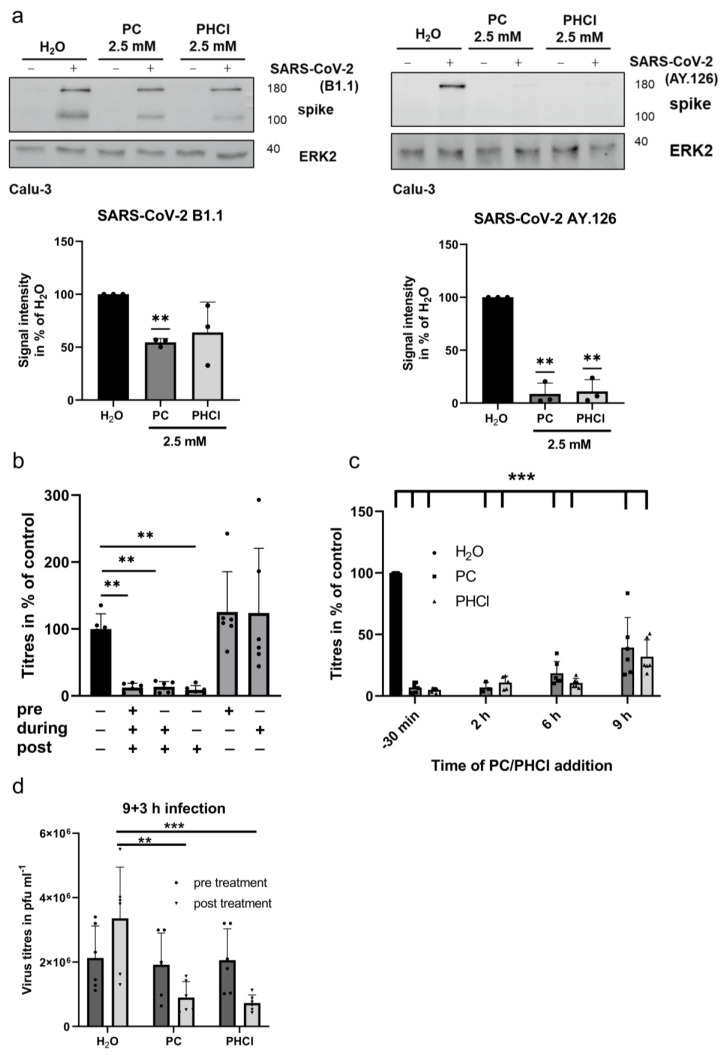
PC and PHCl treatment resulted in reduced SARS-CoV-2 progeny virus production at a late stage of the replication cycle. (**a**) Calu-3 cells were infected with 3 MOI of early SARS-CoV-2 variant or delta variant or were left uninfected in the presence of 2.5 mM PC or PHCl or solvent-treated for 2 h. The medium was replaced, and the cells were further incubated with PC, PHCl or solvent until 8 h p.i., after which cells were lysed and the spike protein content was analysed via SDS-PAGE and Western blot. A representative example of three independent experiments is depicted (top row). Quantification of three independent experiments using ERK2 as loading control was performed using Image J (bottom row). The signal intensity of the solvent-treated samples was arbitrarily set to 100%. Statistical significance was determined using one-sample t-test and compared to 100%; ** *p* < 0.01. (**b**,**c**) Calu-3 cells were treated with 2.5 mM PC (**b**,**c**) or PHCl (**c**) or solvent control as indicated 30 min prior to infection with 3 MOI SARS-CoV-2 B1.1 variant. The cells were again inhibitor- or solvent-treated as indicated during infection. The medium was changed 2 h p.i. and the inhibitor or solvent was added again at the indicated times post infection without removing the media again. Viral titres in the supernatants 8 h (**b**) or 12 h (**c**) p.i. were determined via plaque assay. The mean (+SD) of three independent experiments with biological duplicates is shown. Titres of the solvent-treated samples were arbitrarily set to 100%. Statistical significance was determined using matched ANOVA with Dunette’s multiple comparisons test (**b**) or ordinary two-way ANOVA (**c**). ** *p* < 0.01; *** *p* < 0.001. (**d**) Calu-3 cells were infected with 3 MOI SARS-CoV-2 B1.1 variant for 2 h, after which cells were washed with PBS and the medium was replaced without adding inhibitor. The supernatant was collected 9 h p.i. and replaced with new medium containing 2.5 mM PC or PHCl or solvent. Supernatants were collected at 12 h p.i., and viral titres in all samples were determined via plaque assay. The mean + SD of three independent experiments is shown. Statistical significance was determined via ordinary one-way ANOVA with Dunette’s multiple comparisons test. ** *p* < 0.01; *** *p* < 0.001.

**Figure 4 ijms-24-14584-f004:**
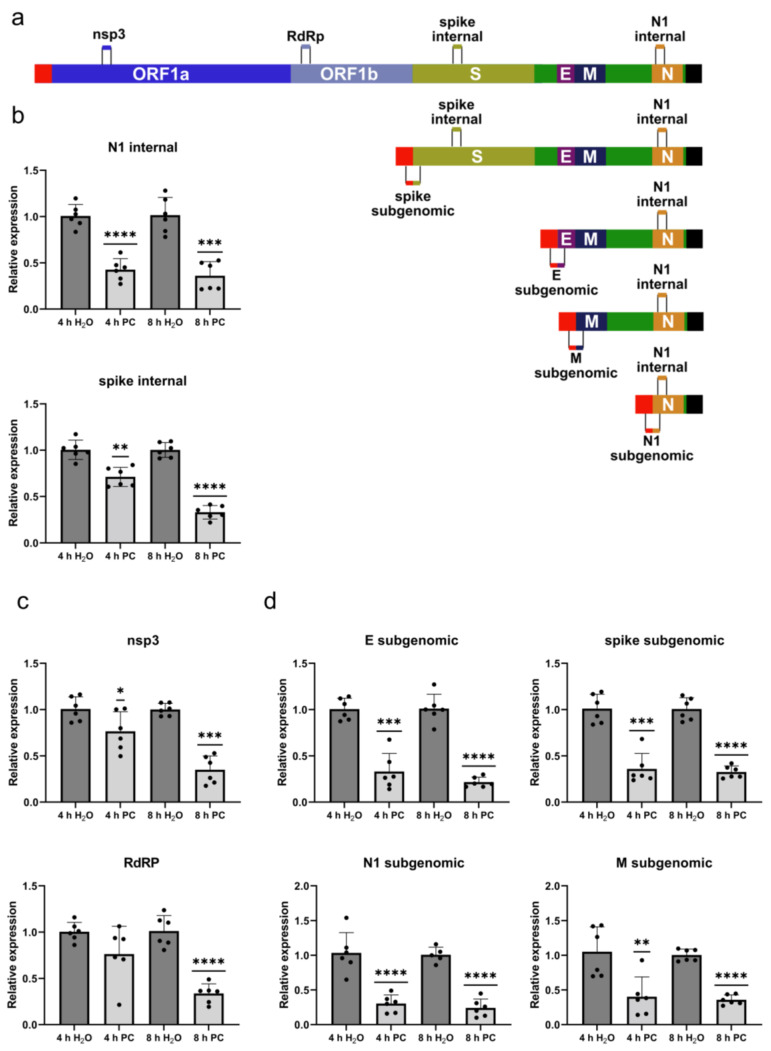
Treatment with PC limits synthesis of subgenomic viral RNAs prior to inhibiting synthesis of whole genome copies. (**a**) Schematic representation of the SARS-CoV-2 genome and the employed primer pairs. (**b**–**d**) Calu-3 cells were infected with 3 MOI of the early SARS-CoV-2 variant for 2 h. The medium was then replaced with new medium containing 2.5 mM PC. The cells were lysed for RNA extraction 4 h and 8 h p.i., and the relative levels of viral RNA were determined via qRT-PCR using primers amplifying (**b**) a structural protein gene region, (**c**) a non-structural protein gene region or (**d**) a region of subgenomic RNA specifically. The mean (+SD) of three independent experiments with biological and technical duplicates is shown. Expression levels of solvent-treated samples were arbitrarily set to 1. Statistical significance was determined via one-sample *t*-test and compared to 1: * *p* < 0.05; ** *p* < 0.01; *** *p* < 0.001; **** *p* < 0.0001.

**Figure 5 ijms-24-14584-f005:**
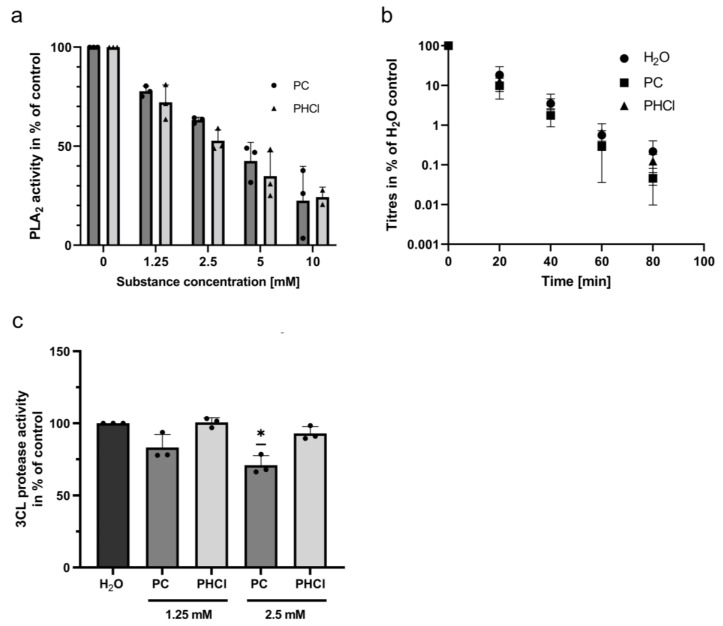
The activity of phospholipase A2 (PLA_2_) from Calu-3 cells is reduced in the presence of PC or PHCl. (**a**) Samples containing PLA_2_ were generated by lysing Calu-3 cells in distilled water. The activity of PLA_2_ in the presence of the indicated concentrations of PC or PHCl was measured using a fluorescent substrate (excitation: 485 nm; emission: 520 nm). PLA_2_ activity of the control sample was arbitrarily set to 100%. The mean + SD of three independent PLA_2_ samples with technical duplicates per treatment are shown. (**b**) Calu-3 cells were infected with 0.5 MOI of SARS-CoV-2 (early variant) for 2 h, after which cells were washed and incubated in new medium containing 2.5 mM PC or PHCl for 24 h. Titres were determined via plaque assay and then equalised in all samples. Samples were subjected to heat treatment at 50 °C for the indicated times, after which titres were again determined via plaque assay. Titres at 0 min were arbitrarily set to 100%. The mean ± SD of three independent experiments with biological duplicates is depicted. (**c**) The activity of SARS-CoV-2 3CL protease in the presence of the indicated concentrations of PC and PHCl was measured using a fluorescent substrate (excitation: 355 nm; emission: 460 nm). Activity in the control samples was arbitrarily set to 100%. The mean + SD of three independent measurements with technical duplicates is shown. Statistical significance was determined via one-sample *t*-test and compared to 100%; * *p* < 0.05.

**Table 1 ijms-24-14584-t001:** List of all primers used.

Name	Sequence	Source
SARS-CoV-2 N1 fw	5′-GACCCCAAAATCAGCGAAAT-3′	
SARS-CoV-2 N1 rv	5′-TCTGGTTACTGCCAGTTGAATCTG-3′	
Human GAPDH fw	5′-CTCTGCTCCTCCTGTTCGAC-3′	
Human GAPDH rv	5′-CAATACGACCAAATCCGTTGAC-3′	
Human IFNβ fw	5′-ATGACCAACAAGTGTCTCCTCC-3′	
Human IFNβ rv	5′-GGAATCCAAGCAAGTTGTAGCTC-3′	
Human IL-6 fw	5′-CAGCCCTGAGAAAGGAGACATG-3′	
Human-IL-6 rv	5′-GCATCCATCTTTTTCAGCCATC-3′	
Human MxA fw	5′-GAAGGGCAACTCCTGACAG-3′	
Human MxA rv	5′-GTTTCCGAAGTGGACATCGCA-3‘	
Human IP10 fw	5′-CCAGAATCGAAGGCCATCAA-3′	
Human IP10 rv	5′-TTTCCTTGCTAACTGCTTTCAG-3´	
Human IFNλ1 fw	5′-CGCCTTGGAAGAGTCACTCA-3′	
Human IFNλ1 rv	5′-GAAGCCTCAGGTCCCAATTC-3′	
Human IFNλ2/3 fw	5′-AGTTCCGGGCCTGTATCCAG-3′	
Human IFNλ2/3 rv	5′-GAGCCGGTACAGCCAATGGT-3′	
Human TNFα fw	5′-GGAGAAGGGTGACCGACTCA-3′	
Human TNFα rv	5′-CTGCCCAGACTCGGCAA-3′	
Human TRAIL fw	5′-GTCTCTCTGTGTGGCTGTAACTTACG-3′	
Human TRAIL rv	5′-AAACAAGCAATGCCACTTTTGG-3′	
Human OAS fw	5′-GATCTCAGAAATACCCCAGCCA-3′	
Human OAS rv	5′-AGCTACCTCGGAAGCACCTT-3′	
Human ACE2 fw	5′-TTCCTGCTCAAACAAGCAC-3′	
Human ACE2 rv	5′-TTCCACCACCCCAACTATC-3′	
Human TMPRSS2 fw	5′-CCTCTAACTGGTGTGATGGCGT-3′	
Human TMPRSS2 rv	5′-TGCCAGGACTTCCTCTGAGATG-3′	
SARS-CoV-2 nsp3 fw	5′-TTCTGCTGCTCTTCAACCTGA-3′	
SARS-CoV-2 nsp3 rv	5′-ATAGTCTGAACAACTGGTGTAAGT-3′	
SARS-CoV-2 RdRp fw	5′-ACGCTCAAAGCTACTGAGGAGAC-3′	OriGene #HP234774
SARS-CoV-2 RdRp rv	5′-GGTCTAGGTTTACCAACTTCCC-3′	
SARS-CoV-2 Spike fw	5′-CAACTGAAATCTATCAGGCCG-3′	OriGene #HP234776
SARS-CoV-2 Spike rv	5′-ACCAACACCATTAGTGGGTTG-3′	
SARS-CoV-2 lead fw	5′-CGATCTCTTGTAGATCTGTTCTC-3′	[53]
Spike subgenomic rv	5′-GAATTAGTGTATGCAGGGGGTAA-3′	
E subgenomic rv	5′-ATATTGCAGCAGTACGCACACA-3′	[53]
M subgenomic rv	5′-CAAATCCATGTAAGGAATAGGAAACC-3′	
N1 subgenomic rv	5′-TCTGGTTACTGCCAGTTGAATCTG-3′	

## Data Availability

The data are contained within the article and Appendix A. The original Western blot pictures were submitted with this article.

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
