# Peer review of "The Local Anaesthetic Procaine Prodrugs ProcCluster® and Procaine Hydrochloride Impair SARS-CoV-2 Replication and Egress In Vitro"

_ijms, 2023, doi:10.3390/ijms241914584_

Round 1

Reviewer 1 Report

In this study Häring et al explore the effects of the anesthetic procaine on SARS-CoV-2 replication in vitro. They show that viral replication was inhibited both early and late during the replication cycle. Intriguingly, the authors show that drug treatment very late (9hpi) in the replication cycle can significantly reduce viral release, which could make procaine a promising antiviral treatment. The study was performed well, and the manuscript is clearly written.

I would request the following changes prior to publication:

All bar graphs should include individual datapoints.

Figure 1C: Please add quantitation of western blots to show that reduced loading control correlates with reduced amount of spike detection

Figure 2: Although I understand the authors’ choice to represent their data as a change from untreated and SARS-CoV-2 infected cells, it reduces the information contained in the data. I would recommend showing the data compared to respective mock controls to showcase the magnitude of the induction of the cytokines, IFN and ISGs.

Figure 5: Was PLA2 activity tested during infection (with/without treatment)? If so, please include the data or discuss this caveat in the discussion.

Discussion: Could the authors please speculate on the differences in the effectiveness of the drug treatments for the alpha and the delta variant infections, especially in relation to the known mutations/differences between the two variants and how this could elude to the mechanism of action for procaine.

Reviewer 2 Report

This is a well conducted study on the effect of procaine prodrugs PC and PHCl on SARS-CoV-2 replication, transcription, release and virus stability from the team led by Christina Ehrhardt . All the conclusions reported by the authors are supported by experimental evidence. I am particularly impressed by the clarity in writing, description of the experimental strategy and results. I have only few comments to for authors to consider. 

1. The higher sensitivity of Delta VOC compared to Alpha is interesting particularly as Delta replicates to higher levels than VOCs and ancestral strain that arose before. In this regard it would be interesting to look at the effect of PC and PHCl on Omicron sublineages as it uses endocytic route to a more extent than PM fusion due to its reduced ability to use TMPRSS2. This experiment could also increase the relevance of the work.

2. Although the authors have clearly shown that SARS-CoV-2 entry is not affected per se, due to the affect on spike protein expression, I would recommend mRNA expression analyses (RT2-PCR ) of entry factors  ACE2 and TMPRSS2 with/without PC/PHCl treatment to strengthen the data.

3. Is it possible for authors to discuss how the inhibitory concentrations (mM)  seen in vitro for PC and PHCl relates to what is used in vivo in patients ? Such a comparison would again help to understand translational potential of the work before use of preclinical animal models.
